# An evaluation of the readability and visual appearance of online patient resources for fibroadenoma

**Hayley Anne Hutchings** *, **Anagha Remesh**

School of Medicine, Faculty of Medicine, Health and Life Science, Swansea University, Swansea, United Kingdom

* h.a.hutchings@swansea.ac.uk

## Abstract

### Introduction

Fibroadenomas are benign lesions found in the breast tissue. Widespread access to and use of the internet has resulted in more individuals using online resources to better understand health conditions, their prognosis and treatment. The aim of this study was to investigate the readability and visual appearance of online patient resources for fibroadenoma.

### Methods

We searched Google™, Bing™ and Yahoo™ on 6 July 2022 using the search terms "fibroadenoma", "breast lumps", "non-cancerous breast lumps", "benign breast lumps" and "benign breast lesions" to identify the top ten websites that appeared on each of the search engines. We excluded advertised websites, links to individual pdf documents and links to blogs/chats. We compiled a complete list of websites identified using the three search engines and the search terms and analysed the content. We only selected pages that were relevant to fibroadenoma. We excluded pages which only contained contact details and no narrative information relating to the condition. We did not assess information where links were directed to alternative websites. We undertook a qualitative visual assessment of each of the websites using a framework of pre-determined key criteria based on the Centers for Medicare and Medicaid Services toolkit. This involved assessing characteristics such as overall design, page layout, font size and colour. Each criterion was scored as: +1- criterion achieved; -1- criterion not achieved; and 0- no evidence, unclear or not applicable (maximum total score 43). We then assessed the readability of each website to determine the UK and US reading age using five different readability tests: Flesch Kincaid, Gunning Fog, Coleman Liau, SMOG, and the Automated Readability Index. We compared the readability scores to determine if there were any significant differences across the websites identified. We also generated scores for the Flesh Reading Ease as well as information about sentence structure (number of syllables per sentence and proportion of words with a high number of syllables) and proportion of people the text was readable to.

**Data Availability Statement:** Data are available from https://zenodo.org/record/6992835#.Yv_Bji7MI2x.

**Funding:** The author(s) received no specific funding for this work.

**Competing interests:** The authors have declared that no competing interests exist.

## Results

We identified 39 websites for readability and visual assessment. The visual assessment scores for the 39 websites identified ranged from -19 to 31 points out of a possible score of 43. The median readability score for the identified websites was 8.58 (age 14–15), with a range of 6.69–12.22 (age 12–13 to university level). There was a statistically significant difference between the readability scores obtained across websites (p<0.001). Almost half of the websites (18/39; 46.2%) were classified as very difficult by the Flesch Reading Ease score, with only 13/39 (33.33%) classified as being fairly easy or plain English.

## Conclusion

We found wide differences in the general appearance, layout and focus of the fibroadenoma websites identified. The readability of most of the websites was also much higher than the recommended level for the public to understand. Fibroadenoma website information needs to be simplified to reduce the use of jargon and specificity to the condition for individuals to better comprehend it. In addition, their visual appearance could be improved by changing the layout and including images and diagrams.

## Introduction

Fibroadenomas are benign breast lesions which are found in the breast tissue. They generally affect premenopausal women and are one of the most common benign tumours of the breast in women under 35 years of age [1–3]. The incidence rate in the adolescent population is 2.2% [4]. These lesions are said to occur in one in four women [5] and account for two-thirds of breast lesions in young women [6].

Generally, there is no need to treat fibroadenomas as they are harmless and rarely lead to malignancy [7]. The options are either to observe the growth or to excise it. Although the latter is uncommon, it is generally based on the lesion size and the healthcare professional's recommendation [8, 9].

There has been a huge increase in the number of people in the last two decades with access to a computer and the ability to search the internet. The development of the world wide web and increased utilisation of the internet has provided an opportunity for individuals to search for health information online whereas previously they would have accessed medical staff [10].

In 2019, it was estimated that 96% of the UK population had access to the internet, giving people access to a vast array of information in the comfort of their own homes [11]. More people than ever are said to be searching the internet for health information for themselves, family and friends [12]. The internet has become the first port of call for many regarding their health [13]. It has been estimated that 37% of internet traffic involves searching for information relating to health conditions [14]. It has been suggested that as people have improved access to online-based health information it is likely that they will become more engaged and involved in decision making [15]. Given the scale of internet searching, it is therefore important that the information provided is accessible, readable and up to date for those accessing it.

The Centers for Disease Control and Prevention (CDC) define Personal Health Literacy as 'the degree to which individuals have the ability to find, understand, and use information and services to inform health-related decisions and actions for themselves and others' [16]. Individuals with low literacy levels may not be able to read a book or newspaper, understand road

signs or price labels, make sense of a bus or train timetable, fill out a form, read instructions on medicines, or use the internet [17]. A high proportion of individuals in the UK and US have been documented as having below average levels of general literacy. The National Literacy Trust has estimated that between 1 in 8 and 1 in 4 adults have a general literacy level below the expected of a UK year 6 student (age 11) [17]. Similar figures have been documented in the US, with 52% of the population having only basic (US 4[th] or 5[th] grade, age 10–11) or below levels of literacy [18]. Table 1 illustrates the readability grades and the corresponding school grades in the UK and US [19].

Fibroadenoma has a relatively high incidence particularly in younger people but has a positive outcome in a high proportion of cases. It is important that the online information available for fibroadenoma is therefore informative and understandable such that individuals can understand the nature of the condition and make informed decisions regarding their treatment choices. In terms of breast care, previous research regarding the readability of online patient resources for breast augmentation and breast cancer have illustrated that online material was above the recommended reading age and more needed to be done to improve the quality and readability of such information [20, 21]. The Ricci et al. study also suggested that high readability could be a barrier to individuals seeking surgery [20].

To our knowledge there has been no previous research undertaken to assess online information for fibroadenoma. The aim of this study was therefore to assess the readability of online information for fibroadenoma and evaluate their visual appearance.

## Materials and methods

We aimed to identify and evaluate websites that contained relevant information about fibroadenoma. We did this in three stages: 1) identification of relevant online websites; 2) qualitative visual assessment of each website; and 3) readability assessment of each website. HAH and AR both did each step, compared the results, and resolved the conflicting results by conversation.

**Table 1. Readability grades with equivalent ages in education in the UK and US [19].**

| Age | Readability Grade | |
|---|---|---|
| | UK | US/Canada |
| 0–2 | - | - |
| 2–3 | Nursery | - |
| 3–4 | Nursery | - |
| 4–5 | Reception | - |
| 5–6 | 1 | Kindergarten |
| 6–7 | 2 | 1 |
| 7–8 | 3 | 2 |
| 8–9 | 4 | 3 |
| 9–10 | 5 | 4 |
| 10–11 | 6 | 5 |
| 11–12 | 7 | 6 |
| 12–13 | 8 | 7 |
| 13–14 | 9 | 8 |
| 14–15 | 10 | 9 |
| 15–16 | 11 | 10 |
| 16–17 | 12 | 11 |
| 17–18 | 13 | 12 |
| 18+ (University or equivalent) | 13–16+ | 12–16+ |

## Identification of online websites

The first stage of the study involved searching and identifying the top websites for fibroadenoma information resources. To establish this, we used the terms "fibroadenoma", "breast lumps", "non-cancerous breast lumps", "benign breast lumps" and "benign breast lesions" and searched Google[TM], Bing[TM] and Yahoo[TM] on 6 July 2022 to identify the top ten websites that appeared on each of the search engines. We chose these three search engines as they are currently the top three used [22].

We excluded advertised websites, links to individual pdf documents and links to blogs/chats. We identified the top 10 sites listed on each of the search engines. We decided to exclude paid advertisements as they receive under 10% of search traffic [23]. We decided to keep Wikipedia in the final list. Although controversial in terms of content quality, Wikipedia is commonly used by the public as a resource, and we therefore considered that it was important to include it.

We compiled a complete list of websites identified using the three search engines. We then analysed the content of each of the identified sites. We aimed to assess up to 10 pages for each of the identified websites. Where fewer pages were available, we analysed the maximum number of relevant pages that were available for that site. We only selected pages that were relevant to fibroadenoma. We excluded pages which only contained contact details and no narrative information relating to the condition. We did not assess information where links were directed to alternative websites. We recorded screenshots and dated them to ensure that we had a permanent record of them on the day the searches were undertaken.

## Qualitative assessment of the identified websites

For the second stage of the study, we undertook a qualitative visual assessment of the identified websites using a pre-determined framework. This was based on an adaption of the guidance provided in the Centers for Medicare and Medicaid Services toolkit (https://www.cms.gov/Outreach-and-Education/Outreach/WrittenMaterialsToolkit) [24, 25].

The CMS government tool kit aims to "provides a detailed and comprehensive set of tools to help you make written material in printed formats easier for people to read, understand, and use"[25]. We considered 43 different variables including the use of different fonts, the distribution of headings and subheading and if the site had a clear path for the eye to follow, to name a few aspects. We scored each of the 43 variables to provide an overall qualitative assessment of the visual aesthetic of the resource. We assigned a score of +1 point if the statement was achieved for each of the 43 variables; 0 if there was no evidence, evidence was unclear or the criterion was not applicable; and -1 point if the statement was not achieved.

## Readability assessment of each website

We assessed readability of the information from each of the identified websites through the Readable website (https://readable.com). Readable is a website which allows for a chosen piece of text to be entered into a website allowing a range of readability formulae to be applied to the text. We entered the information from each page into Readable. We utilised a number of readability formulae to assess the readability of the written content. We used the various readability formulae to directly compare the readability of the text from the identified websites.

We used five of the available readability formulae from Readable to broaden our evaluation of the readability of the websites. Each of these formulae assesses different aspects of the text, and some are specifically used for medical purposes (see Table 2).

For this study we used the following readability formulae: Flesch-Kincaid [26], Gunning Fog [27], Coleman-Liau Index [28], Simple Measure of Gobbledygook Index (SMOG) [29] and the Automated Readability Index [30].

**Table 2. Readability formulae used in the study with their equations.**

| Formula | Equation |
|---|---|
| Flesch Kincaid Grade Index (Kincaid, 1975 [26]) | $= 0.39 \left( \dfrac{total\ words}{total\ sentences} \right) + 11.8 \left( \dfrac{total\ syllables}{total\ words} \right) - 15.59$ |
| Gunning-Fog Index (Gunning, 1952 [27]) | $= 0.4 \times \left[ \left( \dfrac{total\ words}{total\ sentences} \right) + 100 \left( \dfrac{complex\ words}{total\ words} \right) \right]$ |
| Coleman-Liau Index (Coleman, 1975 [28]) | $= (0.0588 \times L) - (0.296 \times S) - 15.8$ |
| Simplified Measure of Gobbledygook (SMOG) (McLaughlin, 1969 [29]) | $= 3 + \sqrt{polysyllabic\ count}$ |
| Automated Readability Index (Smith EA, 1967 [30]) | $= 4.71 \left( \dfrac{characters}{words} \right) + 0.5 \left( \dfrac{words}{sentences} \right) - 21.43$ |

A good readability score should be as low as possible as this classifies that the text is comprehensible for a greater proportion of the public. Readable says that a score of 8 or below indicates 85% of the population are able to comprehend the text in question [31]. This would translate to the reading age of 13–14 years of age.

We standardised the text format prior to calculating the readability score to avoid bias between different formulae. This included removing images, advertisements, and side panels, such as navigation panels, seen on the first page of each website. We left in sub-headings, and bulleted lists as they were in the original website layout.

We calculated five readability scores for each website using each of the five formulae. We calculated a median (and range) readability score for each website. We used the Kruskal-Wallis test to compare the median readability grades across each of the websites identified. We ran a post hoc analysis to determine where these significances were, if applicable. A p value of less than 0.05 was regarded as statistically significant.

In addition to calculating the readability scores, we also generated scores for the Flesh Reading Ease [32] as well as information about sentence structure (number of syllables per sentence and proportion of words with a high number of syllables) and proportion of people the text was readable to.

## Results

### Identification of online websites

We identified 39 websites using the search terms 'fibroadenoma', 'breast lumps', 'non-cancerous breast lumps', 'benign breast lumps' and 'benign breast lesions' using Google™ Yahoo™ and Bing™ search engines. The websites normally occurred across one or two pages of the search engine. There was substantial repetition of sites beyond the top 10, hence we deemed that exploring sites beyond this offered limited additional value. S1 Table in S1 File illustrates the websites identified using the different search terms across the three search engines.

Table 3 illustrates the websites identified (excluding advertisements) along with brief initial appearance assessment and content information.

### Qualitative assessment of the identified websites

The visual assessment undertaken based on the CMS toolkit [25] is shown in S2 Table in S1 File. We identified a large score range indicating high variability between the websites in terms of their visual aesthetics. The highest scores were achieved by the Breast Cancer Now website

**Table 3. Details of websites identified utilising Google™ Bing™ and Yahoo™ search engines.**

| Website Name | Web address | Brief appearance assessment and summary content information |
|---|---|---|
| 1. Breast Cancer Now | https://breastcancernow.org/ https://breastcancernow.org/information-support/have-i-got-breast-cancer/benign-breast-conditions/fibroadenoma | From first appearance, Breast Cancer Now is aesthetically appealing and easy to read with plain colours and good spacing between text. There are clear definitions with any medical jargon being broken down. It also includes a brief outline of symptoms and types of fibroadenoma along with causation, diagnosis, and treatment. This makes it an effective resource for patients. In addition, there are clear diagrams on the page to add to the text which aids understanding of the condition. The website contains general breast care information which is extremely important regarding breast healthcare. There are also many hyperlinks and resources attached to this page which allows the patient to learn more about their condition |
| 2. Mayo Clinic | https://www.mayoclinic.org/ | Mayo Clinic, at first glance, was very minimal with a lot of white space. There were clear divides between each section of the website, although there were very few sections. There were clear definitions as well as symptoms, types, and causations of fibroadenomas being explained very well. There was no information on the diagnosis, treatment, or general breast. Visually, this website was not particularly aesthetic because of lack of diagrams, tables, and charts. The use of advertisements on the page was quite distracting for the reader. Overall, Mayo Clinic, although providing some key basic information was quite limited in terms of the information contained within it. |
| 3. Breast Cancer Organisation | https://www.breastcancer.org/ | The Breast Cancer Organisation website was difficult to read due to the block text nature making the text hard to process. There was no use of highlighting key ideas or bullet pointing lists. Definitions were discussed well though the different types of fibroadenomas were not discussed. The causation and diagnosis were discussed briefly. In addition, there was no mention of general breast care, treatment, or diagrams all of which would have improved the appearance of this resource. One aspect included in this resource was the alterative diagnosis of a lump of a similar geometry to fibroadenomas which was not seen in many other sites |
| 4. National Health Service (NHS) | https://www.nhs.uk/ | The NHS web page was specifically designed to inform regarding breast lumps rather than specifically for fibroadenomas. In general, information on fibroadenomas was limited with only some brief definitions along with causation and diagnosis. From a visual standpoint, this was a clear page with good spacing and plain colours, making the page readable and engaging. In terms of fibroadenomas, there was minimal specific information for the condition. This website however, provided useful links to Breast Cancer Now (website 1). |
| 5. American Cancer Society | https://www.cancer.org/ https://www.cancer.org/cancer/breast-cancer/non-cancerous-breast-conditions/fibroadenomas-of-the-breast.html | The Cancer Organisation website was very well written with the purpose of clearly educating the reader. This could be seen with the detail given when discussing the treatments. The information provided was minimal but of that that was mentioned, it was all beneficial and informative. The lack of images made the website text heavy, making it unappealing to the reader. |
| 6. HealthLine | https://www.healthline.com/ https://www.healthline.com/health/fibroadenoma-breast | From a general perspective, Health Line was very informative in nature with almost all the relevant information regarding fibroadenomas being included in the resource. However, the general layout of the website provided a non-reliable viewpoint with multiple advertisements. This may also be distracting for readers. In terms of content, information regarding the condition, different types and the treatment and diagnosis were all included in the page. The lack of images was noticeable and could have added value to the information provided. Healthline also considered the condition from the patient's perspective when discussing living with a lump and how to deal with this. This enhanced the website however the nature of the site did not allow this to be showcased and patients may not have considered this a reliable source due to the visual appearance. |

*(Continued)*

**Table 3.** (Continued)

| Website Name | Web address | Brief appearance assessment and summary content information |
|---|---|---|
| 7. Medical News Today | https://www.medicalnewstoday.com/ | Medical News Today contained good information regarding fibroadenomas to provide a basic understanding with clear and concise messages for the reader to take away. The website included good information regarding basic definitions, symptoms, diagnosis, and treatments. The different types of fibroadenoma and the causation of them was not included in this patient resource. Images included were of a woman being examined which was good from a visual assessment point of view but could have added more to the content by being complementary to the text. |
| 8. The Women's Hospital | https://www.thewomens.org.au/ | The Woman's Organisation is a hospital website, based in Victoria, Australia where they focus of women's health. The website was very informative containing lots of information regarding the basic definitions, symptoms, diagnosis, and treatments. There were no images which would have aided the website from a visual standpoint. General breast care was explained well here, and this would have been a good opportunity for images to support this information. This, overall, was a good resource whilst being deeply informative for patients with little to no insight into the benign condition. |
| 9. Wikipedia | https://en.wikipedia.org/ | Wikipedia, the world-renowned website, was included as we felt patients would turn to this for insight on the benign condition. In general, the website was very informative. However it was also very detailed with the continual use of medical and scientific jargon which may not be helpful for patients. Additionally, with Wikipedia being freely editable, all information may not be correct and therefore this is another aspect patient must consider when using Wikipedia. Basic ideas of the condition were explained very well including definitions, symptoms, diagnosis tests and treatments. There was no information regarding the different types of fibroadenomas which could occur. There was minimal information regarding general breast care and all images used were mainly cytological meaning they were not beneficial for patient education. In summary, Wikipedia provided a very detailed and informative page on fibroadenomas but unfortunately contained jargon, limiting the accessibility for the public. |
| 10. Teach Me Surgery | https://teachmesurgery.com/ | Teach Me Anatomy is produced as a teaching resource and therefore was pitched at a very high for the public or patients. Overall, this was a clear website from a visual standpoint with key words and terms in bold. Basic ideas of the condition were explained well including definitions, symptoms, and diagnosis tests. However, all other aspects such as treatment and breast care as well as types of fibroadenomas were not mentioned. In addition to this, all the diagrams used were applicable to a clinician with these being medical images and hence less beneficial for the patients. In terms of patient information and education, this would not cater well making it less accessible for readers of all backgrounds. |
| 11. Buoy Health | https://www.buoyhealth.com | Buoy Health overall was a very clear website which was easy to read and lots of bold headings and white space. It was broken down into a few basic categories that covered what fibroadenoma is, the symptoms, causes, treatment and prevention and when to see a doctor. It also included a 'fibroadenoma quiz' to aid people determine whether or not they had fibroadenoma. There were links from the main page to other relevant information. There were no visual images. The website provided a very general overview. |
| 12. Cleveland Clinic | https://my.clevelandclinic.org/ | Cleveland Clinic provided a general overview of fibroadenoma. This included it symptoms and causes, diagnosis and tests, management and treatment, prevention, outlook and prognosis and how to live with fibroadenoma. It contained hyperlinks to other relevant information from the main page. The main page appeared slightly cluttered at the top with information about how to get appointments. The text is clear with a reasonable amount of space. The right-hand panel has space for adverts. No images used. |

*(Continued)*

**Table 3.** (Continued)

| Website Name | Web address | Brief appearance assessment and summary content information |
|---|---|---|
| 13. HCA Healthcare | https://www.hcahealthcare.co.uk/ | HCA Healthcare was a general website that focused mainly on patient appointments. It provided only very brief details about fibroadenoma which included symptoms, diagnosis and potential treatment options. The website was fairly bland with limited use of colour and no images. The start of the webpage was dominated by information about booking an appointment. There were no links to any further information. |
| 14. My Breast My Health | https://mybreastmyhealth.com/ | My breast my health has an engaging website that provides an overview of fibroadenoma. There is brief information provided but it covers what fibroadenoma is, if it is likely to be cancerous and whether follow up is required. It includes some hyperlinks to other relevant information. It includes an image of an ultrasound scan, but it is very difficult to see for the untrained eye. |
| 15. WebMD Cancer Center | https://www.webmd.com/ | WebMD landing page is quite distracting with adverts at the top, to the right and within the main text of the page which distracts from the information. The background page is white which helps with contrast in relation to most of the text. Some of the other contrast text is however very light which may limit readability when printed. The website provides a general overview of fibroadenoma including diagnosis and treatment. It also includes some hyperlinks to other relevant information. There are no figures. |
| 16. Gp notebook | https:/gpnotebook.com/ | GP notebook landing page is immediately distracting with adverts at the top and to the right of the landing page. The text is contrasted on a white background, but the font colour is light blue which could limit readability for some people or when printed. There is very brief information provided but there are links to some reference material. The website is written in a scientific language and appears to be pitched for healthcare professionals. There are no figures. |
| 17. Patient info | https://patient.info/ | The Patient info main landing page contains adverts at the top and to the right and is distracting. It contains general information about breast lumps including a small section on fibroadenoma. There are hyperlinks to other relevant information. There are no images on the main information pages, but lots of images at the bottom of the webpages that link to other articles that are suggested by the site- but that are mostly adverts and external sites. Some of the pages include videos which are a useful resource for patients. Visually the information provided is easy to read. |
| 18. Very well health | https://www.verywellhealth.com/ | Very well health provides information on benign breast lumps including fibroadenomas. Visually the website is appealing and there is good contrast of text and background colour. The website includes adverts, but they are slightly less distracting as they appear only on the right panel, and they are not dynamic. The website includes some figures which help to reinforce the text. The text is reasonably easy to read. The pages include hyperlinks to other relevant information. |
| 19. Bupa UK | https://cms-sc.bupa.co.uk/ | The Bupa website provides clear information about benign breast lumps including brief mention of fibroadenoma. It describes symptoms, diagnosis and how breast lumps can be treated. It does not include external adverts but there are adverts about getting a Bupa appointment. The information is clear and fairly easy to read. It includes links to other relevant information. It also includes a clear figure of the breast. |
| 20. Net Doctor | https://www.netdoctor.co.uk/ | The Net Doctor website is immediately distracting as the reader is immediately presented with dynamic adverts throughout the web pages. The first image is a photograph of a breast covered with a rose, which although pleasant is not relevant to the information provided. The website provides only very basic information about breast lumps with no specific information about fibroadenomas. There are some hyperlinks to other relevant information. |

(*Continued*)

**Table 3.** (Continued)

| Website Name | Web address | Brief appearance assessment and summary content information |
| --- | --- | --- |
| 21. Medicine.net | https://www.medicinenet.com/ | Medicine.net website is immediately distracting due to the number of dynamic and irrelevant adverts. The page is quite difficult to navigate as it contains adverts and often irrelevant information. It does contain hyperlinks to quizzes and slideshows for more information. It contains general information about many types of breast lumps including some brief information on fibroadenomas. The information is largely contained within one long page rather than separate links/tabs which often makes reading difficult. The page begins with a figure of the breast, but the quality is poor, and it is not related to any of the information in the text. |
| 22. Radiologyinfo.org | https://www.radiologyinfo.org/ | Radiology info is defined as a website for patients. Despite this some of the language used is quite technical. The site gives a very general overview of breast lumps, without any specific focus on any one type. There was no reference to fibroadenoma. The main focus of the website appears to be how breast lumps are diagnosed and treated. There is a lot of information of various radiological techniques used to diagnose breast lumps, which is not unexpected given the website name. There are some figures and a slideshow within the main pages and hyperlinks to other relevant information. The look of the website is clear and professional. |
| 23. Stony Brook Cancer Center | https://cancer.stonybrookmedicine.edu/ | The focus of Stony Brook Medical Center appears to be for individuals seeking medical appointments and the start of the website includes information about how to get an appointment. Although there are no external adverts, this initially distracts the reader. The website provides a very brief overview of different types of breast lumps, which includes benign and malignant. Fibroadenomas are not specifically mentioned although fibrocystic changes to the breast are discussed. The website includes some figures which break the text up, but the quality is poor, and they are not described in any way. There is only one page of information and no additional hyperlinks provided for further information. |
| 24. Cancer Research UK | https://www.cancerresearchuk.org/ | The Cancer Research UK site has a focus mainly on breast cancer, although it includes some information on breast lumps. Changes in the breast and procedures used to diagnose breast conditions. It includes links to relevant information, but it is difficult to navigate between pages as each link leads to a number of other sources of information. Visually however the website is clear and uncluttered with appropriate language for the public. The website does not contain any images. |
| 25. John Hopkins Medicine | https://www.hopkinsmedicine.org/ | Visually the Johns Hopkins Medicine site looks clear and professional. Although it is promoting its services it does so in a considerate way, the contact information at the end of the page. The website includes general information on common benign breast conditions and includes some brief information about fibroadenomas. In includes some nice, clear diagrams of the breast. There are links to other relevant information. Some of the information is a little scientific. |
| 26. Nathan T Thomas MD | https://dallas-obgyn.com/ | The focus of the Nathan T Thomas website is to promote services and the top of the webpage includes contact information. The webpage is quite appealing with some figures and contrasting colours. The font sizes are quite small which may make reading difficult for some. There was only one extremely brief page of information about non-cancerous breast lumps and their treatment but no links to further information. Fibroadenomas are not explicitly mentioned, although fibrosis is covered briefly. The figure does not help with describing the information provided. |

(*Continued*)

**Table 3.** (Continued)

| Website Name | Web address | Brief appearance assessment and summary content information |
|---|---|---|
| 27. Memorial Sloan Kettering Cancer Center | https://www.mskcc.org/ | Although focused on promoting services, the website looks very clean, uncluttered, and professional. It includes photographs, lots of white space and contrast as well as hyperlinks to additional information. It includes fairly brief general information about different types of benign breast lumps, including a small section about fibroadenomas. Some of the language is rather scientific. The photographs do not reinforce the text. |
| 28. Iq clinic | https://www.icliniq.com/ | The Iqclinic includes lots of pop-ups, which are quite distracting to the reader. The information is contained within one long page. It includes very brief general information about breast lumps, including a small section on fibroadenomas. Most of the information is in the form of bullet points without and detail given. The website looks rather cluttered and there is limited white space. There are some hyperlinks to further information but no figures. |
| 29. Shape.com | https://www.shape.com/ | The Shape website includes lots of dynamic which are immediately distracting to the reader. The website includes a general overview of the types of benign breast lumps including a small section on fibroadenomas. It includes an informative video which describes the various types of lumps described in the text. It is written in a 'magazine' style and avoids scientific language. It includes some hyperlinks to additional relevant information. |
| 30. The London Clinic | https://www.thelondonclinic.co.uk/ | The London Clinic is immediately distracting due to the pop-up advert that appears. It contains limited overview information about the types of benign breast lumps and how they can be treated. Fibroadenomas are mentioned very briefly. The information is contained all within one page. There are no hyperlinks for further information or figures. Some of the language is rather scientific. |
| 31. Total Health | https://www.totalhealth.co.uk/ | Total health's remit appears to be about attracting patients. Despite this the contact information is fairly well concealed and does not distract from the information provided. The website provides a summary of different benign breast lumps, including a small section on fibroadenomas. The format looks professional with lots of white space. Some of the contrast colours may be difficult to read for some and the font size is small. There are no links to further information, but a glossary is provided that defines some medical/scientific terms. |
| 32. Medanta | https://www.medanta.org/ | Medanta website appears with an advert across the top advertising appointments. The colouring of the website is engaging with an image of the breast at the top. It provides information about distinguishing between benign and malignant breast lesions in terms of symptoms, causes, risks, and prevention. The information is however very limited and often includes bullet points only. Many of the sections require expansion of the text which makes the website rather unwieldy. There are a small number of links to further relevant information. Font size is a little small. |
| 33. Specialist Breast Cancer Surgery | https://www.breastcancerspecialist.com.au/ | The webpage includes information about how to make contact for an appointment at the top pf the page, as well as a contact form to the right, which is slightly distracting. The webpage has an image of a clinician in scrubs, but no specific image relating to breast conditions. The information is well spaced with plenty of white space. The website has only one page of information which summarises the main types of benign breast conditions. There is a small section on fibroadenomas. There are no links to further relevant information. |

(*Continued*)

**Table 3.** (Continued)

| Website Name | Web address | Brief appearance assessment and summary content information |
|---|---|---|
| 34. Breast cancer hub | https://www.breastcancerhub.org/ | The Breast Cancer Hub website includes a banner across the top which is distracting and obscures some of the text. The website includes basic information about benign breast conditions. There is a figure at the top of the page that describes what benign means, but the colours and the font size used make this difficult to read. The information is contained within one page, and this looks very cramped as there is limited use of white space. Some words are underlined, which implies a hyperlink, but it appears that certain terms are underlined for emphasis only. There are some tabs that direct to videos which contain relevant information. |
| 35. Family doctor | https://familydoctor.org/ | This website is immediately distracting as it contains dynamic adverts at the top, to the right and integrated with the text. It includes a very brief summary about benign breast conditions and includes a small section on fibroadenomas. There is an image at the top of the page, but this does not aid undertenant of the text contained within the webpage. There is only one long page of information and no hyperlinks to further relevant information. |
| 36. Moffitt Cancer Center | https://moffitt.org/ | The Moffitt Cancer Center website provides a very brief single page overview about benign breast lumps. It includes information at the top pf the page about how to make an appointment, but this is not too distracting. It includes an image of a 'doctor' showing a patient an x-ray, which does not appear to be specifically relevant to breast lumps. There are some hyperlinks, but these send to a list of information which makes navigation of the website difficult. |
| 37. Up to date | https://www.uptodate.com/ | Up to date is a very text dense website that contains lots of academic information about benign breast lesions, and which includes academic references. A small section is included on fibroadenomas. The language is very scientific. Only one page of information is available to non-subscribers. There are hyperlinks to additional information, but these are only accessible by subscribing. There are no figures, and the organisation is not very appealing for a non-academic. There is limited colour and use of white space. |
| 38. Komen | https://www.komen.org/ | Komen is a charitable organisation, and the focus of the website is on breast cancer research. The main page includes a banner at the top for patients to make contact or for their queries to be answered. The look so the website if professional and it includes some bright engaging colours and some photos, although the photos are not relevant to the breast. It includes information about benign breast conditions. It gives a brief overview of various benign conditions including fibroadenoma. The page is a little cumbersome to use as a lot of the sections require expansion of the text. There are hyperlinks to other relevant information. |
| 39. iheartpathology | https://www.iheartpathology.net/ | The iheartpathology website has a single page summary about benign diseases of the breast. It is very abstract and has lots of bright engaging figures. Its focus is on the pathological aspects of breast disease. Some of the images may be difficult to interpret as a non-clinician and the language is very scientific. Fibroadenoma is mentioned but only in relation to one of the images and it is not really described. There are no hyperlinks to other relevant information. There is limited use of white space, and the appearance is rather cluttered. |

with a score of 31 and the Bupa UK site with a score of 30. The lowest scoring websites were iheartpathology with a score of -19 and Medanta with a score of -6.

Both Breast Cancer Now and Bupa UK were simple and visually appealing (see Table 3). They were well-spaced-out with a good amount of white space. There were no advertisements distracting the reader from the information. The text was well spaced out with bold headings and sub-headings. This helped to break up the text, making the layout of information more

digestible for the reader. The font was consistent throughout both websites, and this reflects positively in the scoring system used for the visual assessment. They both included images which were useful for the target audience.

The lowest scoring website (iheartpathology) appeared engaging initially due to the bright colours and images, but the information contained within the images and text was very scientific and difficult to interpret for a lay person. There was only a single page of information with no links and limited explanation. The website was also very cluttered with limited white space. Medanta had a low visual assessment score due to the inclusion of distracting adverts, limited information, and the unwieldy nature of the website.

### Readability assessment of each website

There was a range in the number of pages on each website providing information regarding fibroadenoma. Some websites contained only a single page, some contained 10 or more pages, whilst others provided somewhere in between. We assessed each of the relevant pages for readability for each website and generated a median score (where possible) for the readability statistics.

We obtained five readability scores for each of the 39 identified websites using Readable (see Table 4). The median readability score for the identified websites was 8.58 (age 14–15), with a range of 6.69–12.22 (age 12–13 to university level). This illustrates the large range of readability scores across the 39 websites.

The websites with the lowest median readability scores were the WebMD Cancer Center and Medanta, which both had a median readability score of 6.69, which equates to a reading age of 12–13 years. The Nathan T Thomas MD and Specialist Breast Cancer Surgery websites had high median readability scores of 12.22 and 12.10 respectively. This equates to a reading age of between 17 years of age to university level.

The Kruskal-Wallis test identified a p value of 0.001 indicating that there was a statistically significant difference in the readability scores across the 39 websites. A post-hoc analysis identified that there were significant differences between the median readability scores of many of the websites (see S3 Table in S1 File).

Additional readability assessment further identified differences between the 39 websites (see Table 4). The Flesh Reading Ease score ranged between 22.14 (Very difficult) for the Gp notebook website to 70.43 (Easy) for the National Health Service website. Almost half of the websites (18/39; 46.2%) were classified as very difficult by the Flesch Reading Ease score, with only 13/39 (33.33%) classified as being fairly easy or plain English.

The proportion each website was readable to ranged between 64% to 85%. Exploration of the number of sentences and words with large numbers of syllables and letters did not reveal any consistent pattern or major differences across the websites.

Fig 1 illustrates both the readability and visual assessment for each of the 39 websites. There was a large amount of variation between the websites for both categories. Some websites performed well in the readability assessment but badly in the visual assessment and vice versa. A low readability score and a high visual assessment score is most desirable.

### Discussion

In general, we found that online resources for fibroadenoma were at a level too high for the public both in terms of their readability and visual appearance. This was seen across all the 39 websites analysed. This highlights that patient resources are not being creating at an accessible and informative level. This may result in miscommunication or misunderstanding of the

**Table 4. Readability assessment of the 39 websites identified on Google™, Yahoo™ and Bing™.**

| Readability Assessments | Websites | | | | | | | | | | | | | | |
|---|---|---|---|---|---|---|---|---|---|---|---|---|---|---|---|
| | 1 | | | 2 | | | 3 | | | 4 | | | 5 | | |
| | Grade (US) | Grade (UK) | Age | Grade (US) | Grade (UK) | Age | Grade (US) | Grade (UK) | Age | Grade (US) | Grade (UK) | Age | Grade (US) | Grade (UK) | Age |
| Flesch-Kincaid | 6.50 | 7.50 | 12–13 | 8.01 | 9.01 | 13–14 | 6.93 | 7.93 | 14–15 | 6.08 | 7.08 | 11–12 | 7.04 | 8.04 | 12–13 |
| Gunning Fog | 8.81 | 9.81 | 14–15 | 9.22 | 10.22 | 14–15 | 8.08 | 9.08 | 13–14 | 8.13 | 9.13 | 13–14 | 6.93 | 7.93 | 12–13 |
| Coleman-Liau | 8.94 | 9.94 | 14–15 | 10.88 | 11.88 | 16–17 | 9.70 | 10.70 | 15–16 | 8.33 | 9.33 | 13–14 | 9.8 | 10.8 | 15–16 |
| SMOG | 9.70 | 10.70 | 15–16 | 10.64 | 11.64 | 16–17 | 9.69 | 10.69 | 15–16 | 9.28 | 10.28 | 14–15 | 9.74 | 10.74 | 15–16 |
| Automated Readability | 6.01 | 7.01 | 12–13 | 7.01 | 8.01 | 12–13 | 6.02 | 7.02 | 11–12 | 5.45 | 6.45 | 10–11 | 6.02 | 7.02 | 11–12 |
| Median readability grade | 8.57 | 9.57 | 14–15 | 9.22 | 10.22 | 13–14 | 8.08 | 9.08 | 13–14 | 8.13 | 9.13 | 13–14 | 7.49 | 8.49 | 12–13 |
| Flesch Reading Ease | 68.14 | Plain English | | 54.03 | Fairly difficult | | 56.37 | Fairly difficult | | 70.43 | Fairly Easy | | 60.7 | Plain English | |
| Sentences> 30 syllables | 13% | | | 11% | | | 12% | | | 19% | | | 10% | | |
| Sentences> 20 syllables | 33% | | | 31% | | | 24% | | | 33% | | | 25% | | |
| Words > 4 syllables | 1% | | | 1% | | | 1% | | | 1% | | | 1% | | |
| Words >12 letters | 0% | | | 0% | | | 1% | | | 0% | | | 0% | | |
| % of general public readable to | 85% | | | 84% | | | 85% | | | 85% | | | 85% | | |

| Readability Assessments | Websites | | | | | | | | | | | | | | |
|---|---|---|---|---|---|---|---|---|---|---|---|---|---|---|---|
| | 6 | | | 7 | | | 8 | | | 9 | | | 10 | | |
| | Grade (US) | Grade (UK) | Age | Grade (US) | Grade (UK) | Age | Grade (US) | Grade (UK) | Age | Grade (US) | Grade (UK) | Age | Grade (US) | Grade (UK) | Age |
| Flesch-Kincaid | 7.72 | 8.82 | 13–14 | 8.34 | 9.34 | 13–14 | 6.34 | 7.34 | 11–12 | 10.04 | 11.04 | 15–16 | 10.42 | 11.42 | 15–16 |
| Gunning Fog | 9.23 | 10.23 | 14–15 | 9.93 | 10.93 | 15–16 | 8.62 | 7.62 | 14–15 | 9.67 | 10.67 | 15–16 | 11.41 | 12.41 | 16–17 |
| Coleman-Liau | 10.33 | 11.33 | 15–16 | 10.98 | 11.98 | 16–17 | 9.84 | 10.84 | 15–16 | 12.86 | 13.86 | 18+ | 13.35 | 14.45 | 18+ |
| SMOG | 10.33 | 11.33 | 15–16 | 10.82 | 11.82 | 16–17 | 9.69 | 10.69 | 15–16 | 11.16 | 12.16 | 16–17 | 12.07 | 13.07 | 18+ |
| Automated Readability | 6.86 | 7.86 | 12–13 | 7.41 | 8.41 | 12–13 | 6.34 | 7.34 | 11–12 | 8.53 | 9.53 | 14–15 | 9.48 | 10.48 | 14–15 |
| Median readability grade | 9.23 | 10.23 | 14–15 | 9.93 | 10.93 | 15–16 | 8.45 | 9.45 | 13–14 | 9.98 | 10.98 | 15–16 | 11.41 | 12.41 | 16–17 |
| Flesch Reading Ease | 57.79 | Fairly difficult | | 53.65 | Fairly difficult | | 66.99 | Plain English | | 35.63 | Difficult | | 39.01 | Difficult | |
| Sentences> 30 syllables | 13% | | | 14% | | | 13% | | | 14% | | | 29% | | |
| Sentences> 20 syllables | 30% | | | 35% | | | 28% | | | 24% | | | 45% | | |
| Words > 4 syllables | 1% | | | 2% | | | 1% | | | 5% | | | 3% | | |
| Words >12 letters | 0% | | | 0% | | | 0% | | | 0% | | | 0% | | |
| % of general public readable to | 85% | | | 83% | | | 85% | | | 72% | | | 70% | | |

| Readability Assessments | Websites | | | | | | | | | | | | | | |
|---|---|---|---|---|---|---|---|---|---|---|---|---|---|---|---|
| | 11 | | | 12 | | | 13 | | | 14 | | | 15 | | |
| | Grade (US) | Grade (UK) | Age | Grade (US) | Grade (UK) | Age | Grade (US) | Grade (UK) | Age | Grade (US) | Grade (UK) | Age | Grade (US) | Grade (UK) | Age |
| Flesch-Kincaid | 6.95 | 7.95 | 12–13 | 7.94 | 8.94 | 13–14 | 7.39 | 8.39 | 12–13 | 6.51 | 7.51 | 12–13 | 6.69 | 7.79 | ’12–13 |
| Gunning Fog | 8.57 | 9.57 | 14–15 | 7.13 | 8.13 | 12–13 | 7.16 | 8.16 | 12–13 | 9.04 | 10.04 | 14–15 | 5.81 | 6.81 | 11–12 |
| Coleman-Liau | 9.80 | 10.80 | 15–16 | 11.89 | 12.89 | 17–18 | 9.92 | 10.92 | 15–16 | 8.34 | 9.34 | 13–14 | 10.15 | 11.15 | 15–16 |
| SMOG | 10.13 | 11.13 | 15–16 | 9.91 | 10.91 | 15–16 | 8.87 | 9.97 | 14–15 | 10.17 | 11.17 | 15–16 | 9.23 | 10.23 | 14–15 |
| Automated Readability | 6.52 | 7.52 | 12–13 | 7.64 | 8.64 | 13–14 | 6.46 | 7.46 | 11–12 | 5.54 | 6.54 | 11–12 | 6.41 | 7.41 | 11–12 |
| Median readability grade | 8.57 | 9.57 | 14–15 | 7.94 | 8.94 | 13–14 | 7.39 | 8.39 | 12–13 | 8.34 | 9.34 | 13–14 | 6.69 | 7.79 | ’12–13 |
| Flesch Reading Ease | 62.55 | Plain English | | 51.08 | Fairly difficult | | 50.80 | Fairly difficult | | 67.88 | Plain English | | 58.16 | Fairly difficult | |
| Sentences> 30 syllables | 14% | | | 5% | | | 6% | | | 12–13 | | | 8% | | |
| Sentences> 20 syllables | 30% | | | 16% | | | 11% | | | 14–15 | | | 17% | | |
| Words > 4 syllables | 1% | | | 1% | | | 4% | | | 13–14 | | | 1% | | |
| Words >12 letters | 0% | | | 0% | | | 0% | | | 15–16 | | | 0% | | |
| % of general public readable to | 85% | | | 85% | | | 85% | | | 85% | | | 85% | | |

*(Continued)*

**Table 4.** (Continued)

| Readability Assessments | Websites | | | | | | | | | | | | | | |
|---|---|---|---|---|---|---|---|---|---|---|---|---|---|---|---|
| | 1 | | | 2 | | | 3 | | | 4 | | | 5 | | |
| | Grade (US) | Grade (UK) | Age | Grade (US) | Grade (UK) | Age | Grade (US) | Grade (UK) | Age | Grade (US) | Grade (UK) | Age | Grade (US) | Grade (UK) | Age |

| Readability Assessments | Websites | | | | | | | | | | | | | | |
|---|---|---|---|---|---|---|---|---|---|---|---|---|---|---|---|
| | 16 | | | 17 | | | 18 | | | 19 | | | 20 | | |
| | Grade (US) | Grade (UK) | Age | Grade (US) | Grade (UK) | Age | Grade (US) | Grade (UK) | Age | Grade (US) | Grade (UK) | Age | Grade (US) | Grade (UK) | Age |
| Flesch-Kincaid | 11.36 | 12.36 | 16–17 | 6.88 | 7.88 | 12–13 | 8.05 | 9.05 | 13–14 | 6.24 | 7.24 | 11–12 | 7.13 | 8.13 | 12–13 |
| Gunning Fog | 10.11 | 11.11 | 15–16 | 8.93 | 9.93 | 14–15 | 9.61 | 10.61 | 15–16 | 8.65 | 9.65 | 14–15 | 8.21 | 9.21 | 13–14 |
| Coleman-Liau | 13.83 | 14.83 | 18+ | 8.58 | 9.58 | 14–15 | 10.21 | 11.21 | 15–16 | 8.8 | 9.8 | 14–15 | 10.25 | 11.25 | 15–16 |
| SMOG | 10.47 | 11.47 | 15–16 | 10.28 | 11.28 | 15–16 | 10.94 | 11.94 | 16–17 | 9.72 | 10.72 | 15–16 | 10.31 | 11.31 | 15–16 |
| Automated Readability | 9.66 | 10.66 | 15–16 | 5.26 | 6.26 | 10–11 | 7.1 | 8.1 | 12–13 | 5.47 | 6.47 | 10–11 | 7.09 | 8.09 | 12–13 |
| Median readability grade | 10.53 | | 16–17 | 8.58 | 9.58 | 14–15 | 9.6 | 10.6 | 15–16 | 8.21 | 9.21 | 13–14 | 8.21 | 9.21 | 13–14 |
| Flesch Reading Ease | 22.14 | Very difficult | | 62.41 | Plain English | | 56.9 | Fairly difficult | | 67.57 | Plain English | | 63.21 | Plain English | |
| Sentences> 30 syllables | 9% | | | 9% | | | 18% | | | 11% | | | 17% | | |
| Sentences> 20 syllables | 15% | | | 26% | | | 35% | | | 29% | | | 32% | | |
| Words > 4 syllables | 7% | | | 1% | | | 1% | | | 1% | | | 1% | | |
| Words >12 letters | 0% | | | 0% | | | 0% | | | 0% | | | 0% | | |
| % of general public readable to | 64% | | | 85% | | | 85% | | | 85% | | | 85% | | |

| Readability Assessments | Websites | | | | | | | | | | | | | | |
|---|---|---|---|---|---|---|---|---|---|---|---|---|---|---|---|
| | 21 | | | 22 | | | 23 | | | 24 | | | 25 | | |
| | Grade (US) | Grade (UK) | Age | Grade (US) | Grade (UK) | Age | Grade (US) | Grade (UK) | Age | Grade (US) | Grade (UK) | Age | Grade (US) | Grade (UK) | Age |
| Flesch-Kincaid | 9.09 | 10.09 | 14–15 | 9.53 | 10.53 | 15–16 | 9.53 | 10.53 | 14–15 | 5.82 | 6.82 | 11–12 | 7.58 | 8.58 | 13–14 |
| Gunning Fog | 11.04 | 12.04 | 16–17 | 12.12 | 13.12 | 17–18 | 10.12 | 11.12 | 15–16 | 7.24 | 8.24 | 12–13 | 8.72 | 9.72 | 14–15 |
| Coleman-Liau | 11.24 | 12.24 | 16–17 | 11.24 | 12.24 | 16–17 | 12.8 | 13.8 | 18+ | 9.02 | 10.02 | 14–15 | 11.00 | 12.00 | 16–17 |
| SMOG | 11.75 | 12.75 | 17–18 | 12.28 | 13.28 | 17–18 | 11.94 | 12.94 | 17–18 | 8.85 | 9.85 | 14–15 | 10.50 | 11.50 | 16.17 |
| Automated Readability | 8.32 | 9.32 | 13–14 | 8.4 | 9.4 | 13–14 | 10.04 | 11.04 | 15–16 | 5.51 | 6.51 | 11–12 | 7.16 | 8.16 | 12–13 |
| Median readability grade | 10.77 | 11.77 | 16.17 | 11.24 | 12.24 | 16–17 | 10.12 | 11.12 | 15–16 | 7.24 | 8.24 | 12–13 | 8.72 | 9.72 | 14–15 |
| Flesch Reading Ease | 53.52 | Fairly difficult | | 48.75 | Difficult | | 49.7 | Difficult | | 68.73 | Plain English | | 56.94 | Fairly difficult | |
| Sentences> 30 syllables | 26% | | | 25% | | | 30% | | | 5% | | | 10% | | |
| Sentences> 20 syllables | 45% | | | 47% | | | 48% | | | 19% | | | 25% | | |
| Words > 4 syllables | 2% | | | 2% | | | 2% | | | 1% | | | 3% | | |
| Words >12 letters | 0% | | | 0% | | | 0% | | | 0% | | | 0% | | |
| % of general public readable to | 79% | | | 76% | | | 76% | | | 85% | | | 85% | | |

| Readability Assessments | Websites | | | | | | | | | | | | | | |
|---|---|---|---|---|---|---|---|---|---|---|---|---|---|---|---|
| | 26 | | | 27 | | | 28 | | | 29 | | | 30 | | |
| | Grade (US) | Grade (UK) | Age | Grade (US) | Grade (UK) | Age | Grade (US) | Grade (UK) | Age | Grade (US) | Grade (UK) | Age | Grade (US) | Grade (UK) | Age |
| Flesch-Kincaid | 9.14 | 9.14 | 14–15 | 6.61 | 7.61 | 12–13 | 8.06 | 9.06 | 13.14 | 8.86 | 9.86 | 14–15 | 7.41 | 8.41 | 12–13 |
| Gunning Fog | 12.33 | 13.33 | 17–18 | 7.89 | 8.89 | 13–14 | 8.75 | 9.75 | 14–15 | 10.49 | 11.49 | 15–16 | 8.62 | 9.62 | 14–15 |
| Coleman-Liau | 12.49 | 13.49 | 17–18 | 9.37 | 10.37 | 14–15 | 10.18 | 11.18 | 15–16 | 10.77 | 11.77 | 16–17 | 11.26 | 12.26 | 16.17 |
| SMOG | 12.22 | 13.22 | 17–18 | 9.80 | 10.80 | 15–16 | 10.46 | 11.46 | 15–16 | 11.94 | 12.94 | 17–18 | 9.85 | 10.85 | 15–16 |
| Automated Readability | 10.09 | 11.09 | 15–16 | 5.98 | 6.98 | 11–12 | 6.60 | 7.60 | 12–13 | 8.90 | 9.90 | 14–15 | 7.22 | 8.22 | 12–13 |
| Median readability grade | 12.22 | 13.22 | 17–18 | 7.89 | 8.89 | 13–14 | 8.75 | 9.75 | 14–15 | 10.06 | 11.06 | 15–16 | 8.62 | 9.62 | 14–15 |
| Flesch Reading Ease | 55.22 | Fairly difficult | | 64.28 | Plain English | | 52.19 | Fairly difficult | | 57.87 | Fairly difficult | | 56.32 | Fairly difficult | |
| Sentences> 30 syllables | 20% | | | 13% | | | 10% | | | 33% | | | 10% | | |
| Sentences> 20 syllables | 53% | | | 29% | | | 27% | | | 47% | | | 29% | | |
| Words > 4 syllables | 2% | | | 1% | | | 2% | | | 1% | | | 2% | | |

*(Continued)*

**Table 4.** (Continued)

| Readability Assessments | Websites | | | | | | | | | | | | | | |
|---|---|---|---|---|---|---|---|---|---|---|---|---|---|---|---|
| | 1 | | | 2 | | | 3 | | | 4 | | | 5 | | |
| | Grade (US) | Grade (UK) | Age | Grade (US) | Grade (UK) | Age | Grade (US) | Grade (UK) | Age | Grade (US) | Grade (UK) | Age | Grade (US) | Grade (UK) | Age |
| Words >12 letters | 0% | | | 0% | | | 0% | | | 0% | | | 0% | | |
| % of general public readable to | 77% | | | 85% | | | 84% | | | 80% | | | 85% | | |

| Readability Assessments | Websites | | | | | | | | | | | | | | |
|---|---|---|---|---|---|---|---|---|---|---|---|---|---|---|---|
| | 31 | | | 32 | | | 33 | | | 34 | | | 35 | | |
| | Grade (US) | Grade (UK) | Age | Grade (US) | Grade (UK) | Age | Grade (US) | Grade (UK) | Age | Grade (US) | Grade (UK) | Age | Grade (US) | Grade (UK) | Age |
| Flesch-Kincaid | 7.16 | 8.16 | 12–13 | 6.69 | 7.69 | 12–13 | 10.62 | 11.62 | 16–17 | 9.02 | 10.02 | 14–15 | 6.99 | 7.99 | 12–13 |
| Gunning Fog | 9.28 | 10.28 | 14–15 | 5.82 | 6.82 | 11–12 | 13.06 | 14.06 | 18+ | 10.51 | 11.51 | 16–17 | 8.26 | 9.26 | 13–14 |
| Coleman-Liau | 8.45 | 9.45 | 13–14 | 9.17 | 10.17 | 14–15 | 12.10 | 13.10 | 17–18 | 10.79 | 11.79 | 16–17 | 10.47 | 11.47 | 15–16 |
| SMOG | 10.24 | 11.24 | 15–16 | 9.26 | 10.26 | 14–15 | 12.94 | 13.94 | 18+ | 11.65 | 12.65 | 17–18 | 10.20 | 11.20 | 15–16 |
| Automated Readability | 6.25 | 7.25 | 11–12 | 5.52 | 6.52 | 11–12 | 9.82 | 10.82 | 15–16 | 8.71 | 9.71 | 14–15 | 6.80 | 7.80 | 12–13 |
| Median readability grade | 8.45 | 9.45 | 13–14 | 6.69 | 7.69 | 12–13 | 12.1 | 13.1 | 17–18 | 10.51 | 11.51 | 16.17 | 8.26 | 9.26 | 13–14 |
| Flesch Reading Ease | 66.39 | Plain English | | 57.91 | Fairly Difficult | | 44.81 | Difficult | | 55.96 | Fairly difficult | | 61.42 | Plain English | |
| Sentences> 30 syllables | 19% | | | 5% | | | 44% | | | 30% | | | 7% | | |
| Sentences> 20 syllables | 42% | | | 15% | | | 56% | | | 47% | | | 22% | | |
| Words > 4 syllables | 1% | | | 1% | | | 3% | | | 1% | | | 1% | | |
| Words >12 letters | 0% | | | 0% | | | 0% | | | 0% | | | 0% | | |
| % of general public readable to | 85% | | | 85% | | | 68% | | | 78% | | | 85% | | |

| Readability Assessments | Websites | | | | | | | | | | | |
|---|---|---|---|---|---|---|---|---|---|---|---|---|
| | 36 | | | 37 | | | 38 | | | 39 | | |
| | Grade (US) | Grade (UK) | Age | Grade (US) | Grade (UK) | Age | Grade (US) | Grade (UK) | Age | Grade (US) | Grade (UK) | Age |
| Flesch-Kincaid | 7.68 | 8.68 | 13–14 | 9.16 | 10.16 | 14–15 | 7.82 | 8.82 | 13–14 | 8.58 | 9.58 | 14–15 |
| Gunning Fog | 7.17 | 8.17 | 12–13 | 8.83 | 9.83 | 14–15 | 8.28 | 9.28 | 13–14 | 10.37 | 11.37 | 15–16 |
| Coleman-Liau | 10.69 | 11.69 | 16–17 | 10.31 | 11.31 | 15–16 | 10.1 | 11.1 | 15–16 | 12.72 | 13.72 | 18+ |
| SMOG | 9.78 | 9.78 | 15–16 | 10.87 | 11.87 | 16–17 | 9.57 | 10.57 | 15–16 | 10.78 | 11.78 | 16–17 |
| Automated Readability | 6.68 | 7.68 | 12–13 | 6.42 | 7.42 | 11–12 | 6.34 | 7.34 | 11–12 | 8.42 | 9.42 | 13–14 |
| Median readability grade | 7.68 | 8.68 | 13–14 | 9.16 | 10.16 | 14–15 | 8.28 | 9.28 | 13–14 | 10.37 | 11.37 | 15–16 |
| Flesch Reading Ease | 51.89 | Fairly difficult | | 42.96 | Difficult | | 55.33 | Fairly difficult | | 48.03 | Difficult | |
| Sentences> 30 syllables | 14% | | | 13% | | | 7% | | | 17% | | |
| Sentences> 20 syllables | 16% | | | 22% | | | 19% | | | 29% | | |
| Words > 4 syllables | 1% | | | 2% | | | 1% | | | 4% | | |
| Words >12 letters | 0% | | | 0% | | | 0% | | | 0% | | |
| % of general public readable to | 85% | | | 77% | | | 85% | | | 82% | | |

content currently provided to patients, which may have detrimental effects on physician associated trust or compliance with physician advice.

Readable states that readability scores of 8 or below indicate that the written material is comprehensible for 85% of the population [31]. This score was only achieved by eight of the 39 websites. This suggests that the fibroadenoma resources are not catering and educating the public as we would hope. Similarly a Flesh Reading Ease score lower than 60 is classed as fairly difficult [32], yet only 13/39 (33.33%) identified were classified as being fairly easy or plain English.

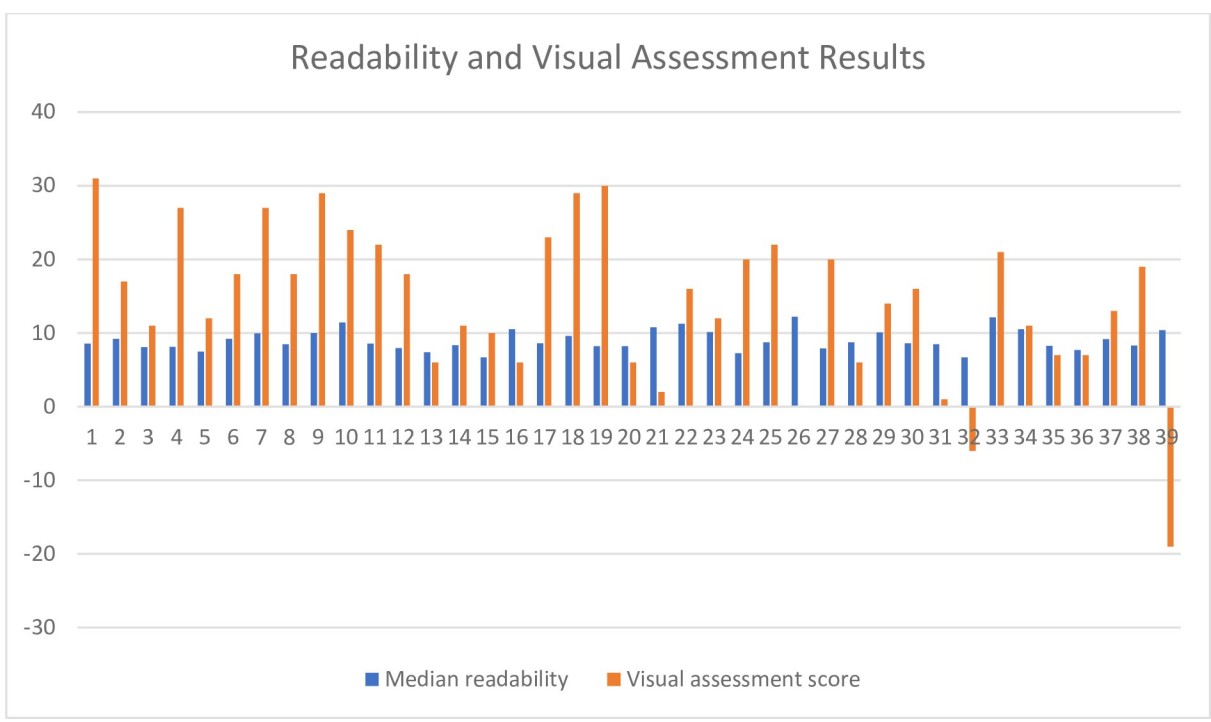

**Fig 1. Readability and visual assessment of the 39 identified websites.**

These finding correlate with the findings from other readability research based on online patient resources for Phenylketonuria and Skin Cancer treatments [19, 33]. These online resources were also pitched at a level too high for members of the public to understand. Similar findings were also seen for online resources for breast cancer and breast augmentation [20, 21].

The findings suggest that website developers need to consider the content they post to ensure that it is clear and accessible to their target audience. Our findings point to a number of ways in which the websites could be improved. Word and sentence length is important as shorter words and sentences are thought to be more comprehensible. The use of simpler vocabulary and less subject specific jargon and clear information written in a concise manner are also preferable. Defining new or complex words and making them stand out may also help readers [34]. By ensuring text is comprehensible to a larger proportion of the public, this will enable better public health education. Secondary and community healthcare providers could help support patients by developing their own resources that have a low readability score and that are visually engaging [35]. They should also consider directing patients to websites that are trusted, provide understandable information and that are engaging to patients. If these were hosted on healthcare providers websites, this would reassure patients about their credibility.

In terms of visual assessment, there are various ways in which websites can be improved to achieve a higher score. The Centers for Medicare and Medicaid Services toolkit [25] is a useful resource to inform website development. The visual image of the first page of the website is important as this is the first interaction the user will have with it, even before reading the written content. It is therefore important to make sure there is a clear and obvious path for the eye to follow. This includes the removal of advertisements and unnecessary images and creating a

clean and uncluttered layout. Ensuring that the page is appealing at first glance also includes the use of colour sparingly. When considering colour usage, it is important to maintain that the text is still easy to read. The best way to achieve this is by using a dark coloured font on a light-coloured background. It is also important to consider if the colour scheme chosen also works if printed.

The layout of the text is also an important consideration which can improve the visual assessment score. Text size should be large enough to be easily read without changing the view of the page and an easy-to-read font should be used [24, 36]. The UK Government suggests that public resources should be written in Arial or Helvetica font [37]. Ways of improving the visual appearance also include: the use of bold or highlighted key words, bullet pointed lists, spacing between lines of text and paragraphs, and the use of headings and subheadings. This assists with breaking up large blocks of text which may not entice the reader to work their way through the text.

Utilisation of audio-visual materials within websites may also improve accessibility. Our findings illustrated that websites that used simple figures were visually much better than those without. Visual materials can be an effective way to convey and present information in a clear, organized way. In the health setting, the use of audio-visual materials alongside written materials has been shown to be beneficial [38].

There are some limitations of this study. Firstly, the identified websites were only analysed for readability and visual appearance. The readability formulae examine the readability of the text only and are based primarily on information such as the sentence length, number of words per sentence and the complexity of some of the words. Additional website resources such as tables, diagrams and videos may aid the understanding of the information presented on the websites [39] and these were not specifically examined. The readability grades found in our study therefore only give an insight into the overall website readability. We did however visually assess the overall visual appearance of each website which will also play a part in engaging with the patient and similarly found that many of the websites identified performed poorly in terms of visual aesthetics. There are other aspects that could be evaluated to assess the resources. This could include the usefulness of the written content with specific focuses on the scientific knowledge or the quality of the information provided.

By assessing up to a maximum of ten pages on each website it is likely that the median readability score is truly representative of the website. However, some websites only had limited information or a small number of pages and this may have resulted in an overly high readability score. It is important to ensure that home or landing pages are visually engaging and have a low readability score as 90% of users do not progress beyond the first page of an internet search [23]. The first page of each website should be informative enough to provide some basic understanding for a patient with little to no understanding of the condition.

This study only looked at the top ten webpages using Google[TM], Yahoo[TM] and Bing[TM] search engines utilising specific search terms. Although these search engines are the top three most popular and widely used [22], they may not necessarily be the only source of information for patients. Further exploration utilising different search engines, and varying assessment criteria and exploring links to other online resources such as videos and pdf leaflets is necessary.

## Conclusion

Following the introduction of the Internet, more and more people are using it to access information regarding their health. Given the low average literacy levels in the UK and US, it is important that the information presented within websites is accessible and comprehensible. We found that available resources for fibroadenoma are above the recommended reading age

and that for most the visual appearance of these resources was poor. This may mean that the fibroadenoma resources are not accessible and educating the public as we would hope.

## Supporting information

**S1 File.**
(DOCX)

## Author Contributions

**Conceptualization:** Hayley Anne Hutchings.

**Data curation:** Anagha Remesh.

**Formal analysis:** Anagha Remesh.

**Investigation:** Anagha Remesh.

**Methodology:** Hayley Anne Hutchings.

**Project administration:** Anagha Remesh.

**Resources:** Hayley Anne Hutchings.

**Supervision:** Hayley Anne Hutchings.

**Validation:** Hayley Anne Hutchings.

**Writing – original draft:** Hayley Anne Hutchings.

**Writing – review & editing:** Hayley Anne Hutchings, Anagha Remesh.

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
