## [Decision Letter · Decision Letter 0]

17 Jun 2022

PONE-D-21-37567An Evaluation of the Readability and Visual Appearance of Online Patient Resources for FibroadenomaPLOS ONE

Dear Dr. Hutchings,

Thank you for submitting your manuscript to PLOS ONE. After careful consideration, we feel that it has merit but does not fully meet PLOS ONE’s publication criteria as it currently stands. Therefore, we invite you to submit a revised version of the manuscript that addresses the points raised during the review process. Your manuscript has been assessed by two expert reviewers, whose comments are appended to this letter. Based on these reports, our major concern with this study is the extent to which the sample of pages you investigated represents the full breadth of online resources available to patients. As reviewer 1 has noted, your study design used a small sample of 10 sites, taken from searching a single search engine using a single keyword.  Please ensure you respond in detail to these concerns, and the other points raised by the reviewers, in your response to reviewers document, and modify your manuscript accordingly.

We look forward to receiving your revised manuscript.

Kind regards,

Joseph Donlan

Editorial Office

PLOS ONE

Journal Requirements:

4. Please upload a copy of Figure 6, to which you refer in your text on page 11. If the figure is no longer to be included as part of the submission please remove all reference to it within the text.

Reviewers' comments:

Reviewer's Responses to Questions

**Comments to the Author**

1. Is the manuscript technically sound, and do the data support the conclusions?

Reviewer #1: Partly

Reviewer #2: Yes

2. Has the statistical analysis been performed appropriately and rigorously? 

Reviewer #1: Yes

Reviewer #2: Yes

3. Have the authors made all data underlying the findings in their manuscript fully available?

Reviewer #1: Yes

Reviewer #2: Yes

4. Is the manuscript presented in an intelligible fashion and written in standard English?

Reviewer #1: Yes

Reviewer #2: Yes

5. Review Comments to the Author

Reviewer #1: Although the study is well executed and on an important topic, the sample size of 10 websites evaluated in 2020 calls into question the scope and strength, as well as the currency, of the authors' findings. Work that evaluated a greater number of websites covering more search phrases than the single "fibroadenoma," perhaps employing more sophisticated analytic techniques to account for greater sample size, or perhaps including resources in other languages, would perhaps be more appropriate for PLOS.

Reviewer #2: I would remove some redundant information and will make the introduction shortened. The introduction includes too detailed information about fibroadenoma that is not relevant to this research purpose, so I would more focus on why assessing the readability of online information for fibroadenoma is important.

I think that two separate people did the readability assessment, compared the results, and resolved the conflicting results by conversation; however, this process is not included in the Methods. I would include such information in the Method.

In the discussion, I would include what type of support can be provided to increase the readability of information in a health care setting and/or community level. The current discussion focused on educating website developers but we may need to think what other level of setting can do to increase its readability.

6. PLOS authors have the option to publish the peer review history of their article (what does this mean?). If published, this will include your full peer review and any attached files.

Reviewer #1: No

Reviewer #2: No

---

## [Author Response · Author response to Decision Letter 0]

19 Aug 2022

Journal Requirements:

Response: We have reformatted our paper based on PLOS ONE’s style requirements.

Response: There are no restrictions in terms of data sharing. We have now deposited our data in the Zenodo opendata repository at Swansea University (Available at: An Evaluation of the Readability and Visual Appearance of Online Patient Resources for Fibroadenoma | Zenodo)

Response: Data have now been deposited on Zenoda (available at: An Evaluation of the Readability and Visual Appearance of Online Patient Resources for Fibroadenoma | Zenodo)

4. Please upload a copy of Figure 6, to which you refer in your text on page 11. If the figure is no longer to be included as part of the submission please remove all reference to it within the text.

Response: Apologies this is an error. All references to Table 6 have been removed. This table is now referred to as Table 4 in the text.

Reviewers' comments:

Reviewer's Responses to Questions

Reviewer comments

Review Comments to the Author

Reviewer #1: Although the study is well executed and on an important topic, the sample size of 10 websites evaluated in 2020 calls into question the scope and strength, as well as the currency, of the authors' findings. Work that evaluated a greater number of websites covering more search phrases than the single "fibroadenoma," perhaps employing more sophisticated analytic techniques to account for greater sample size, or perhaps including resources in other languages, would perhaps be more appropriate for PLOS.

Response: Thank you for these helpful comments to improve the paper. We have now undertaken a more extensive search and analysis of the data: 

1. We used three search engines- Google, Bing and Yahoo to identify relevant websites.

2. We expanded the search terms used with these three search engines:

a. Fibroadenoma

b. Breast lumps

c. Non-cancerous breast lumps

d. Benign breast lumps

e. Benign breast lesions

3. We have now used more extensive assessment metrics to evaluate the readability of the 39 identified websites.

Reviewer #2: I would remove some redundant information and will make the introduction shortened. The introduction includes too detailed information about fibroadenoma that is not relevant to this research purpose, so I would more focus on why assessing the readability of online information for fibroadenoma is important.

Response: Thank you. We have now shortened the introduction and removed some redundant information. The Introduction is now focused on readability. 

I think that two separate people did the readability assessment, compared the results, and resolved the conflicting results by conversation; however, this process is not included in the Methods. I would include such information in the Method.

Response: Thank you. This is correct- we have now included this information in the Methods.

In the discussion, I would include what type of support can be provided to increase the readability of information in a health care setting and/or community level. The current discussion focused on educating website developers but we may need to think what other level of setting can do to increase its readability.

Response: Thank you. We have now included information in the discussion about this.

---

## [Decision Letter · Decision Letter 1]

4 Nov 2022

An Evaluation of the Readability and Visual Appearance of Online Patient Resources for Fibroadenoma

PONE-D-21-37567R1

Dear Dr. Hutchings,

We’re pleased to inform you that your manuscript has been judged scientifically suitable for publication and will be formally accepted for publication once it meets all outstanding technical requirements.

Kind regards,

Miquel Vall-llosera Camps

Senior Editor

PLOS ONE

Reviewers' comments:

Reviewer's Responses to Questions

**Comments to the Author**

1. If the authors have adequately addressed your comments raised in a previous round of review and you feel that this manuscript is now acceptable for publication, you may indicate that here to bypass the “Comments to the Author” section, enter your conflict of interest statement in the “Confidential to Editor” section, and submit your "Accept" recommendation.

Reviewer #1: All comments have been addressed

2. Is the manuscript technically sound, and do the data support the conclusions?

Reviewer #1: Yes

3. Has the statistical analysis been performed appropriately and rigorously? 

Reviewer #1: Yes

4. Have the authors made all data underlying the findings in their manuscript fully available?

Reviewer #1: Yes

5. Is the manuscript presented in an intelligible fashion and written in standard English?

Reviewer #1: Yes

6. Review Comments to the Author

Reviewer #1: Thank you for addressing my comments and suggestions. I think with the revision, the paper is much improved.

7. PLOS authors have the option to publish the peer review history of their article (what does this mean?). If published, this will include your full peer review and any attached files.

Reviewer #1: **Yes: **Edward Christopher Dee

---

## [Editor Report · Acceptance letter]

9 Nov 2022

PONE-D-21-37567R1 

An Evaluation of the Readability and Visual Appearance of Online Patient Resources for Fibroadenoma 

Dear Dr. Hutchings:

I'm pleased to inform you that your manuscript has been deemed suitable for publication in PLOS ONE. Congratulations! Your manuscript is now with our production department. 

Kind regards, 

on behalf of

Dr. Miquel Vall-llosera Camps 

Staff Editor

PLOS ONE